# Using a Bayesian network to classify time to return to sport based on football injury epidemiological data

Kate K. Y. Yung[ID][1,2,3,4]*, Paul P. Y. Wu[2,3], Karen aus der Fünten[4], Anne Hecksteden[5,6], Tim Meyer[4]

1 Department of Orthopaedics and Traumatology, Faculty of Medicine, The Chinese University of Hong Kong, Shatin, Hong Kong, 2 School of Mathematical Sciences, Queensland University of Technology, Brisbane, Queensland, Australia, 3 Centre for Data Science, Queensland University of Technology, Brisbane, Queensland, Australia, 4 Institute of Sports and Preventive Medicine, Saarland University, Saarbrücken, Germany, 5 Institute of Sports Science, University of Innsbruck, Innsbruck, Austria, 6 Institute of Physiology, Medical University of Innsbruck, Innsbruck, Austria

* kate.yung@cuhk.edu.hk

## Abstract

The return-to-sport (RTS) process is multifaceted and complex, as multiple variables may interact and influence the time to RTS. These variables include intrinsic factors related the player, such as anthropometrics and playing position, or extrinsic factors, such as competitive pressure. Providing an individualised estimation of time to return to play is often challenging, and clinical decision support tools are not common in sports medicine. This study uses epidemiological data to demonstrate a Bayesian Network (BN). We applied a BN that integrated clinical, non-clinical factors, and expert knowledge to classify time day to RTS and injury severity (minimal, mild, moderate and severe) for individual players. Retrospective injury data of 3374 player seasons and 6143 time-loss injuries from seven seasons of the professional German football league (Bundesliga, 2014/2015 through 2020/2021) were collected from public databases and media resources. A total of twelve variables from three categories (player's characteristics and anthropometrics, match information and injury information) were included. The response variables were 1) days to RTS (1–3, 4–7, 8–14, 15–28, 29–60, >60, and 2) injury severity (minimal, mild, moderate, and severe). The sensitivity of the model for days to RTS was 0.24–0.97, while for severity categories it was 0.73–1.00. The user's accuracy of the model for days to RTS was 0.52–0.83, while for severity categories, it was 0.67–1.00. The BN can help to integrate different data types to model the probability of an outcome, such as days to return to sport. In our study, the BN may support coaches and players in 1) predicting days to RTS given an injury, 2) team planning via assessment of scenarios based on players' characteristics and injury risk, and 3) understanding the relationships between injury risk factors and RTS. This study demonstrates the how a Bayesian network may aid clinical decision making for RTS.

**Data availability statement:** Dataset cannot be shared publicly because the data used for this study were collected, summarised and verified from various media outlets/databases and therefore include patient privacy. At this level of play (Bundesliga 1), effective annoymisation is also not possible. Due to above reason and to protect patient privacy, Saarland University does not allow the data to be shared publicly. Data are available from the from the Institute for Sports and Preventive Medicine, Saarland University (contact via sportmed@mx.unisaarland.de) for researchers who meet the criteria to access to confidential data.

**Funding:** The author(s) received no specific funding for this work.

**Competing interests:** The authors have declared that no competing interests exist.

# Introduction

## Challenges in return to sport decision making

Return to sport (RTS) is defined as when an injured athlete can return to full unrestricted team training and play without modifications in duration and/ or activities[1–3]. Forecasting or estimating the return date of an injured athlete is crucial for team planning, performance optimisation, and game strategy development. By having an estimated return date, medical staff can create an individualised rehabilitation plan to support a graduated progression in the athlete's conditioning program to improve the athlete's condition and assist in mitigating re-injury risk. The provision of predicted timeframe for recovery also helps coaching staff to adjust their game plans and strategies, maximising the team's chances of success.

Athletes operate within a complex system and are influenced by factors such injury history, current injury, body mass index, playing positions, sociological factors, psychological status, and the nature of the sports event [4,5]. Accurately predicting an athlete's RTS can be a challenging task for medical staff due to the intricate interactions between various variables and their influence on injuries. Understanding the interactions between the human physiological system's and injury risk is crucial to provide accurate injury prognoses and project an athlete's likely RTS time range (e.g., there is a 75% chance for the player to return between 4–5 weeks). Such RTS projections could help medical staff and athletes to prepare RTS programmes based on both the lower and upper projection limits.

Large-scale epidemiological studies have estimated the expected RTS time for major injuries in football at the population level accounting for associated injury risk factors[6–10]. These studies describe the risk factors for a disease or injury and the extent of the problem. Medical staff can then use an "anchor and adjust" strategy to optimise predictive accuracy, thus better estimating RTS times[11]. While typical epidemiological studies aim to estimate population-level effect and trends, tools to translate population-level data into individual-level estimations would be helpful.

Forecasting the time to RTS based on individual characteristics can be challenging for several reasons. First, the *Strategic Assessment of Risk and Risk Tolerance Framework* recommends considering personal characteristics and factors, such as age and playing position[12]. However, quantifying and synthesising this information in the clinical reasoning process can be difficult. For example, what is the difference between the rehabilitation time required for a 20-year-old striker versus a 30-year-old defender with a same type of hamstring strain injury? Second, synthesising useful information from broad population-level epidemiological data covering entire leagues and multiple seasons and then personalising it to a specific player-context is almost cognitively infeasible for a single individual or even a small group of individuals. There are no tools to help translate the sheer volume and complexity of the population-level data and tailor them to an individual level. Third, medical staff, like any other humans, are susceptible to various decision-making challenges. They have limited information processing capacity[13] and face potential cognitive biases[14,15]. Human judgment can be vulnerable, particularly when it comes to statistical and probabilistic reasoning[16]. Without an objective and reliable decision-making tool specifically designed to provide individualised RTS estimations, medical staff often rely heavily on their clinical experience and population-level epidemiological data.

Data scientists have the potential to develop decision support tools that can effectively synthesise broad epidemiological data and personalise it to specific player-contexts, overcoming the cognitive limitations faced by individuals. These tools can provide valuable additional information that would otherwise be challenging for a single human to process. In practical settings, data scientists can undertake the setting up and maintaining the decision support tool, while users such as medical staff and coaches utilise the tool in planning for an athlete's

RTS, making necessary adjustments for the team, and ensuring the team remains competitive. This collaboration between data scientists and end users can lead to more informed decision-making and strategic planning.

Developing a computer-based decision support tool that uses a complex systems approach may be helpful to overcome the previous challenges and offer a competitive advantage in forecasting RTS[17]. Specifically, the key differentiator of a complex systems approach in RTS is the explicit modelling of factor interactions that could be queried and used by medical staff in multiple ways. This includes, but is not limited to understanding the strength of influence of different variables and predicting the outcome based on custom RTS scenarios.

Computer-based decision support tools can be divided into predictive and descriptive modelling[18]. Predictive modelling can be used for injury diagnosis, severity estimation, and rehabilitation planning. In particular, the Bayesian network (BN) is well suited for providing injury prognosis in sports, due to its capacity to model complex systems and to integrate clinical data, non-clinical factors and expert knowledge. However, it has not been previously used for such a purpose.

## About the Bayesian network

A BN is a graph-based modelling method where the relationships between variables (nodes) are represented with arrows (arcs) (See Fig 1). The presence of an arc denotes the influence of one node on another, and the absence of one assumes conditional independence[19]. While real data often contains a mixture of discrete and continuous variables, BN structure learning algorithms often assume the random variables are discrete. This type of BN is called the discrete BN, which involves discretising continuous variables in the dataset into categories. Although some information is lost when continuous data are categorised[20], there are merits of using the discrete Bayesian network that are worth discussing.

In practical settings, people often find it easier to work with discrete representations rather than continuous data. Discrete variables tend to be more interpretable, facilitating abstract reasoning[21]. For example, word tokens often enable fast and exact processing[21]. Consider the comparison between the words "tall" and "short" versus the numerical values "183cm" and "150cm". Some may find the descriptive terms are easier to grasp and apply in reasoning. Given the complexity of the human body and interacting processes involved in RTS, discretisation may help capture the resolution of the data available and its relevance to the decision support scenario at hand[22].

However, it's also important to note the disadvantages to discretising continuous variables[23]. Discretisation can lead to information loss, as the finer details and nuances captured by continuous data may be overlooked. By converting continuous variables into discrete categories, we sacrifice the precision and granularity inherent in the original data. Additionally, the choice of how to discretise the variables can introduce subjectivity and bias into the analysis and therefore, it is crucial to ensure transparency in how variables are discretised to mitigate these potential issues.

BNs provide a platform for inferring state probabilities given observations, referred to as evidence, of one or more nodes in the network. In the discrete BN, the relationship between parent nodes and a child node can be quantified using conditional probability tables (CPTs). The CPT reflects the probability of child node states (or outcomes) given every possible combination of parent node states[24]. As new evidence comes in, changes may be made to the node's marginal probability[25], which is known as the posterior probability. The posterior probability is the updated probability of a hypothesis or event after considering new evidence. They combine prior beliefs with the likelihood of the data to provide an updated estimate of the event's probability.

BNs can perform both predictive and diagnostic inference. For example, medical staff can use the former to predict the outcome of an injury for a given clinical diagnosis, anthropometric and match factors (predictive inference); but can also enter the injury severity as an observation to examine what injury factors could explain that observation (diagnostic inference). As an important feature to end users who may not be familiar with statistics, the BN provides visuals to facilitate understanding and supporting decision making across teams of users, such as, among athletes, coaching staff, medical and non-medical personnel.

Another key feature of a BN is that medical staff can integrate and visualise data from multiple sources into a single BN, such as clinical data, empirical evidence and expert knowledge[26], or combinations of these[27]. Expert knowledge can be invaluable when empirical data is scarce or unavailable. It also plays a significant role in developing models, selecting data or variables, estimating parameters, interpreting results and determining the uncertainty characteristics. Since the BN can be customised based on the experts' knowledge[28,29], it may be appealing for small sample size research in elite sports research[30–32], where some data may be missing or not be feasible to collect (e.g., limitations in applied settings)[26]. BNs have been used to support clinical decisions[33–36], analyse complex systems in ecology [22,37,38] and logistics[39].

## Study objectives

The objective of this study was to demonstrate the use of a Bayesian network and its potential utility in becoming a decision support tool for medical staff for RTS. Specifically, a discrete BN was modelled based on a set of basic epidemiological data to demonstrate how medical staff can use BN to understand 1) the most influential variable to the outcomes, 2) the strength of influence of different variables, and to 3) demonstrate its use with a case scenario.

## Materials and methods

### Study design

The study is a retrospective analysis of prospectively collected injury data from the German professional men's highest football league (Bundesliga) between 2014/2015–2020/2021. Neither research ethics board approval nor a trial registration was required as all data were collected from publicly available sources[9,40]. We reported the result with reference to the Transparent Reporting of a multivariable prediction model for Individual Prognosis or Diagnosis (TRIPOD) statement[41].

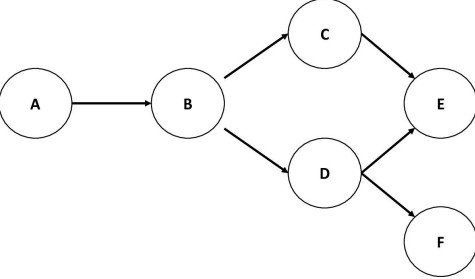

**Fig 1. A graph-based modelling showing nodes (A to F) and arcs (arrows).** A is the parent of B, while B is the child of A; B is the parent of C and D, while C and D are the children of B; and so on. A is the ancestor of B, C, D, E and F, while B, C, D, E and F are the descendants of A; and so on. The arrows indicate the direction of influence.

## Participants

Players who played in the Bundesliga in the above-mentioned seven seasons were included in the study. Injured players who did not return to the Bundesliga in the same season and those without complete data were excluded from the analysis. All participants were identified using a publicly available database, including Kicker Sportmagazin™, and clubs' official websites. Data collection was performed via methods established in previous investigations[9,42]. The data has been collected according to the Fuller et al. consensus statement for football injury research and the same definitions have been the same throughout the 7 seasons[43]. 3374 player Bundesliga seasons were registered over the seven seasons from 2014/2015–2020/2021. A total of 6653 time-loss injuries were recorded. After removing injuries without complete data, 6143 time-loss injuries remained.

## Construction of the Bayesian network

The modelling process and validation were performed in GeNIe 2.0 (Bayes Fusion, LLC)[44]. There are four main steps to creating and validating a BN:

1) Identify the variables (nodes) of the BN and the key response variables. In this study, the key response variables are the (i) severity of the injury *(severity)* and (ii) days to RTS *(days_rts)*. They are used to evaluate the performance of the BN. The nodes and sources of information are outlined in Table 1.

2) Learn the graphical structure of the BN and display the relationships between all the nodes.

3) Learn the probability distribution governing the relationships between the nodes.

4) Validate the BN to evaluate how well the BN can classify injury based on the key response variables: (i) *severity* and (ii) *days_rts*.

**Step 1: Identify the main variables.** This model is intended to capture the conditional probabilistic relationships among personal characteristics, match and injury information. The variables in this study are commonly included in epidemiologic studies[45] and we have summarised them in Table 1.

All time-loss injuries that occurred during football competitions and training sessions were included, and the day of the injury is counted as day zero. A time-loss injury is when a player cannot fully participate in training or competition due to injury[43]. RTS is defined as when the player has received medical clearance to allow full participation in training and is available

**Table 1. Summary of data type and source of information.**

| Category | Variable *(node name in the BN)* | Source |
|---|---|---|
| Personal characteristics and anthropometrics | • Age *(age)*<br>• Height *(height)*<br>• Weight *(weight)*<br>• Main playing position *(main_position)* | Kicker Sportmagazin™ and clubs' official website |
| Match information | • Time of the season *(time_season)*<br>• Training or match *(training_game)* | Ligainsider, official team websites, injury reports, official team press releases and professional statistical websites. |
| Injury information | • Type of injury *(type_injury)*<br>• Body region *(body_region)*<br>• Contact/non-contact *(contact_noncontact)*<br>• Severity *(severity)*<br>• Days to RTS *(days_rts)* | |

for match selection[2]. Details of the injury (type of injury, body region, contact/non-contact injury, training or match) were labelled based on the Fuller et al. consensus of data collection in football[43]. The time of the season was based on an injury epidemiologic study in European football[46]. Personal characteristics and anthropometrics, match information and the above information are used as the explanatory variables for constructing the model.

We are interested in determining the days to RTS, therefore, the two key response variables in this model are injury severity (***severity)*** and days to RTS ***(days_rts)***. Injury severity (*severity*) was categorised according to the days of absence in match or training as outlined by the Fuller et al. consensus statement on football injury studies[43]: minimal (1–3 days), mild (4–7 days), moderate (8–28 days), or severe (>28 days). We further create a new variable based on days to RTS *(days_rts)* to evaluate if the model could classify the days to RTS into more precise categories. There are two additional categories in *days_rts* as compared to *severity*: 1–3, 4–7, 8–14, **15–28**, 29–60, >**60** days. In context, these variables form the nodes in the BN. A summary of the variables and source of the data are presented in Table 1.

**Step 2: Define the graphical structure.** We incorporated expert knowledge to constrain the search to help ensure graph structures produced are consistent with clinical science (See S1 Fig). We first specified the temporal order of the variables, ensuring there were no arcs from variables that occurred later (e.g., injury) to nodes happening earlier (e.g., main playing position). Following this, Author KY established the relationships (arcs) within the graphical structure. Notably, the key response variable, *days_rts*, was directly linked to *severity* in the graphical structure. This mapping was based on the definition of severity outlined in the Fuller et al. consensus statement on football injury studies, which defines severity by the duration of unavailability for full training[43]. This mapping was done manually based on clinical knowledge and could not be learnt from algorithms. Finally, with the constraints in place, we used the Bayesian Search algorithm in GeNIe 2.0 to find the best-fitted network for the optimal network configuration that aligns with the collected data and expert knowledge[19,47].

**Step 3: Define the probability distribution.** There are multiple ways to discretise continuous data, including manual, unsupervised and supervised. Each method offers distinct advantages, such as improved model performance, easier interpretation, and computational efficiency[48]. Manual discretisation involves manually defining thresholds or categories to discretise the continuous data. This approach provides flexibility and allows domain experts to incorporate their knowledge, ensuring that the categories align with relevant domain-specific consensus statements and the specific needs of the analysis. Unsupervised discretisation involves using clustering algorithms to identify natural groupings or patterns in the data. These clusters can then be treated as discrete categories. This is particularly useful when prior knowledge or predefined categories are unavailable[49]. Supervised discretisation utilises labelled data to guide the discretisation process. Machine learning algorithms or decision trees can be employed to learn optimal thresholds or categories that maximise predictive performance. This method can enhance both model performance and interpretability[49].

We manually discretised the continuous data based on domain-specific decision categories to maximise usability across various practitioner types and clubs. Specifically, the continuous data are discretised based on categories that are easy to apply practically or with reference to relevant consensus statements. We discretised *age, height,* and *weight* into three categories that represents the typical values observed in the sample, to create uniform count categories that were both meaningful and easy to interpret. *severity* was discretised based on the Fuller et al. consensus of epidemiological data collection into four categories (minimal, mild, moderate and severe)[43]. *days_rts* was discretised into six categories: 1–3, 4–7, 8–14, 15–28, 29–60, >60 days, with reference to the Fuller et al. consensus of epidemiological data collection[43].

Descriptions of the nodes and categories are summarised in Table 2. We used the Expectation Maximisation (EM) algorithm in GeNIe 2.0 to determine the probability distribution (parameter learning) of the dataset[50,51].

**Step 4: Validation of the BN.** A crucial element of learning is to validate the model. Validation was performed on the two target nodes, i.e., s*everity* and *days_rts*, which are the main outcomes of interest. Ten-fold cross-validation was performed where the dataset was split into ten parts of equal probability[52]. The model was trained on nine parts and tested on the remaining tenth part of the unseen data (holdout test sets). The process was repeated ten times, with a different part of the data being used for testing. The model evaluation technique implemented in GeNIe 2.0 keeps the model structure fixed and re-learns the model parameters during each of the folds. We compiled the results of the test splits to report the sensitivity and user's accuracy[53]. The sensitivity is calculated by dividing the total number of correctly classified injury severity by the total number of actual occurrences of injury of that severity and represents the true positive rate for each category.

$$\text{Sensitivity} = \frac{\text{True positive}}{\text{True positive} + \text{False negative}} \quad (1)$$

The user's accuracy is calculated by dividing the total number of correctly classified injury severity by the total classified true occurrences of injury of that severity. User's accuracy provides insights into the precision of the model's predictions for each category and how reliable it is from the users' perspective.

$$\mathbf{User's\,accuracy} = \frac{\mathbf{True\,positive}}{\mathbf{True\,positive} + \mathbf{False\,positive}} \quad (2)$$

**Table 2. Description of data set variables.**

| | Node name | Description | Categories (States) |
|---|---|---|---|
| 1 | *age* | Age at the start of the season. | <24, 24–27,>27 |
| 2 | *height* | Height at the start of the season (cm) | <180,180–186,>186 |
| 3 | *weight* | Weight at the start of the season (kg) | <75, 75–81,>81 |
| 4 | *bmi* | Body mass index (BMI) | <22.8, 22.8–23.8,>23.8 |
| 5 | *main_position* | Playing location in the field | Goalkeeper, defender, midfield and attacker |
| 6 | *body_part* | The body part which was injured | Head/face, neck/cervical spine, shoulder/clavicular, upper arm, elbow, forearm, wrist, hand/finger/thumb, sternum/ribs/upper back, abdomen, lower back/ pelvis/ sacrum, hip/ groin, thigh, knee, lower leg/Achilles tendon, ankle, foot/toe |
| 7 | *type_injury* | The type of injury that occurred | Fractures and bone stress, joint and ligament, muscles and tendon, haematoma/contusions/bruise, laceration and skin lesion, central/peripheral nervous systems, other injuries[1] (Injuries grouped as 'other' include bursitis, peritonitis, capsular tears, chondral lesion, with no individual category accounting to more than 1% of the injuries.) |
| 8 | *contact_non contact* | Whether the injury occurred was contact or non-contact | Contact, non-contact |
| 9 | *training_match* | Whether the injury occurred during a competition or training session | match, training |
| 10 | *time_season* | Part of the season in which the injury occurred | Preseason (Jul–Aug), fall (Sep–Nov), Winter (Dec–Feb), spring (Mar–May) |
| 11 | *days_RTS* | Days to return to competition or training session (days) | 1–3, 4–7, 8–14, 15–28, 29–60, >60 |
| 12 | *severity* | The severity of the injury | Minimal (1–3 days), mild (4–7 days), moderate (8–28 days), severe (>28 days) |

By considering both the sensitivity and user's accuracy, we can evaluate the model's performance in terms of ability to classify correctly and accuracy of predictions within each category. This is particularly relevant when dealing with models that have multiple categories, such as our current model's injury severity classification.

# Results

## Demographics

The demographics and the main playing positions of injured players are shown in S1 Table. The breakdown of injury by nature of injury (contact or non-contact), event (match or training), body region, and types of injuries are available in S2 Table.

## Bayesian network

The network and the probability distribution of each variable are presented in Fig 2. The model's sensitivity in classifying *days_rts* and *severity* is presented in Table 3. In terms of categorising *days_rts*, the sensitivity ranges from 0.24 to 0.97, with the best performance for shorter days (i.e., below 3 days) and the worst performance for the mid-range category (i.e., 8–14 days). In classifying the injury's severity, the sensitivity ranges from 0.73 to 1.00, with the best performance for severe and the worst for minimal. In terms of categorising *days_rts*, the user's accuracy ranges from 0.52 to 0.83, with the best performance for days 3–7 and the worst performance for the mid-range category (i.e., 8–14 and 15–28 days). In classifying the injury's severity, the sensitivity ranges from 0.67 to 1.00, with > 0.90 in all categories except minimal.

In our model, the type of injury and the injured body region are directly connected to *days_rts* (Fig 2). The percentages indicate the distribution of values under each variable, with the blue arrows indicate how factors influence one another (Fig 2). Based on the sensitivity analysis, the type of injury and the injured body region are most influential to the *days_rts*, followed by age, contact or non-contact injury and the nature of the event (training or match) (see Fig 2). Time of the season, weight, and height only had a minor influence on the result. The time of the season is also associated with the nature of the event (training or match) and the *type of injury*, which is supported by empirical evidence[54].

## Sensitivity analysis, feature selection and strength of influence

A sensitivity analysis was performed using built-in functionality in GeNIe 2.0 to determine the influence of the individual nodes. Sensitivity analysis helps determine the influence of observing the states of specific nodes (i.e., prior and conditional probabilities) on the output variables (i.e., posterior probabilities)[55], which in this case, are the days to RTS and severity of the injury (Fig 3). This can help to support the selection of key variables to be included in a model (feature selection). Results of the sensitivity analysis are visually summarised in the form of tornado charts in S2 Fig.

Red-coloured nodes: contain variables important for calculating posterior probability distributions in *days_rts*. Key: A darker red colour indicates a higher degree of influence. Grey-coloured nodes: the node has no influence on the posterior probability distributions of *days_rts*.

Highly sensitive variables affect the inference results more significantly. Identifying the highly sensitive variables directs medical staff to specific areas to focus on to affect the BN's outcome. As an example, in Fig 3, we have set *days_rts* as the key response variable. Nodes coloured in red contain variables important for calculating posterior probability distributions in *days_rts*. Grey-coloured nodes have no influence on the posterior probability distributions

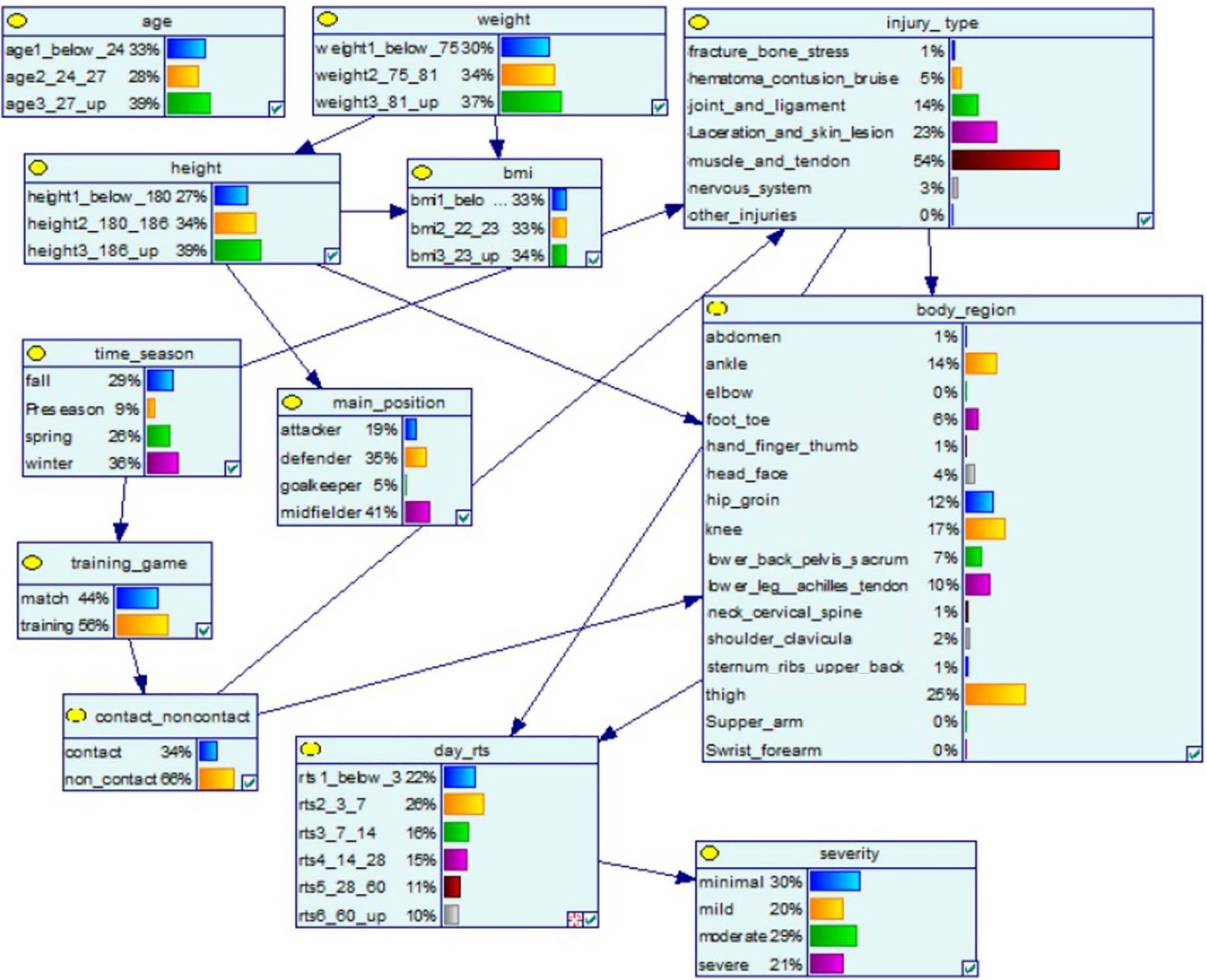

**Fig 2. Structure and network probability of the Bayesian network.** The percentages indicate the distribution of values under each variable, while the blue arrows indicate how factors influence one another.

of *days_rts*, as determined by data and knowledge. Based on the sensitivity analysis, *body_region*, *injury_type* and *contact_noncontact* are the most influential to the outcome in *days_rts* (Fig 3). A numeric form of the strength of influence of different variables can be found in S3 Table.

## Discussion

This study demonstrates the applicability of a BN in an epidemiological dataset to project rehabilitation timelines and provides evidence-based support to RTS decisions. We integrated personal characteristics and anthropometrics, match information and injury information to construct a BN that may inform the days to RTS and injury severity.

**Table 3.** The model's sensitivity's and user's accuracy of classifying days to RTS and severity of the injuries.

| Days to RTS categories | Number of occurrences | Sensitivity | User's accuracy |
|---|---|---|---|
| Below 3 | 1403 | 0.97 | 0.73 |
| Days 3–7 | 1576 | 0.67 | 0.83 |
| Days 8–14 | 1042 | 0.24 | 0.52 |
| Days 15–28 | 888 | 0.76 | 0.52 |
| Days 29–60 | 686 | 0.59 | 0.66 |
| 60 and above | 548 | 0.61 | 0.57 |
| Overall | 6143 | 0.66 | 0.66 |
| **Severity categories** | **Number of Occurrences** | **Sensitivity** | **Users' accuracy** |
| Minimal | 1235 | 0.73 | 1.00 |
| Mild | 1930 | 0.85 | 0.67 |
| Moderate | 1773 | 0.98 | 0.90 |
| Severe | 1205 | 1.00 | 0.98 |
| Overall | 6143 | 0.88 | 0.88 |

## The BN in the context of RTS

In this model, we use a hybrid approach in constructing the BN; that is, we have combined clinical knowledge and data-driven learning (Bayesian Search algorithm) when constructing the graphical structure. This is because constructing the BN using pure algorithmic approaches (i.e., unsupervised learning) can sometimes produce graph structures with unreasonable arcs, such as the model attempting to explain age with severity. However, if we use a pure domain approach, we may miss some of the patterns and linkages between variables that were not observed by clinicians. Therefore, we opted to use a hybrid approach when constructing the BN.

The BN model may help medical staff working in the Bundesliga to view athletes and injuries with a complex systems approach. For example, the model captured the complex relationships between the time of the season, the injury type and injury occurrence (Fig 2). Specifically, in our BN model, the time of the season correlates with the weather, which is then associated with the ground condition [54] and injury occurrence[56,57]. As a key feature of a complex systems approach, the BN can explicitly model the factor interactions/ relationships and query in multiple ways, such as comparing the outcomes of custom RTS scenarios in the section titled Illustrative application of the BN), understanding the most influential factors (in the section titled Sensitivity analysis, feature selection and strength of influence) and the strength of influence (in the section titled Sensitivity analysis, feature selection and strength of influence).

To our knowledge, no study in the Bundesliga has studied the correlations of the above factors, so we use an example from the English Premier League (EPL) to illustrate the possible correlations. In the EPL, the ground condition tends to be drier in the preseason. Warm, dry and hard surfaces may be associated with higher injury occurrence[56,57], possibly due to a higher level of shoe-surface traction influence [58] and faster running speed[59]. On the contrary, wet and muddy ground is associated with lower injury occurrence[57], possibly due to changes in playing style (e.g., less high-speed tackles) and reduced shoe-surface traction. While the time of the season affects the injury occurrence, there has been no direct effect on days to RTS. Analysing these complex relationships would be difficult without the use of computers and advanced statistical modelling.

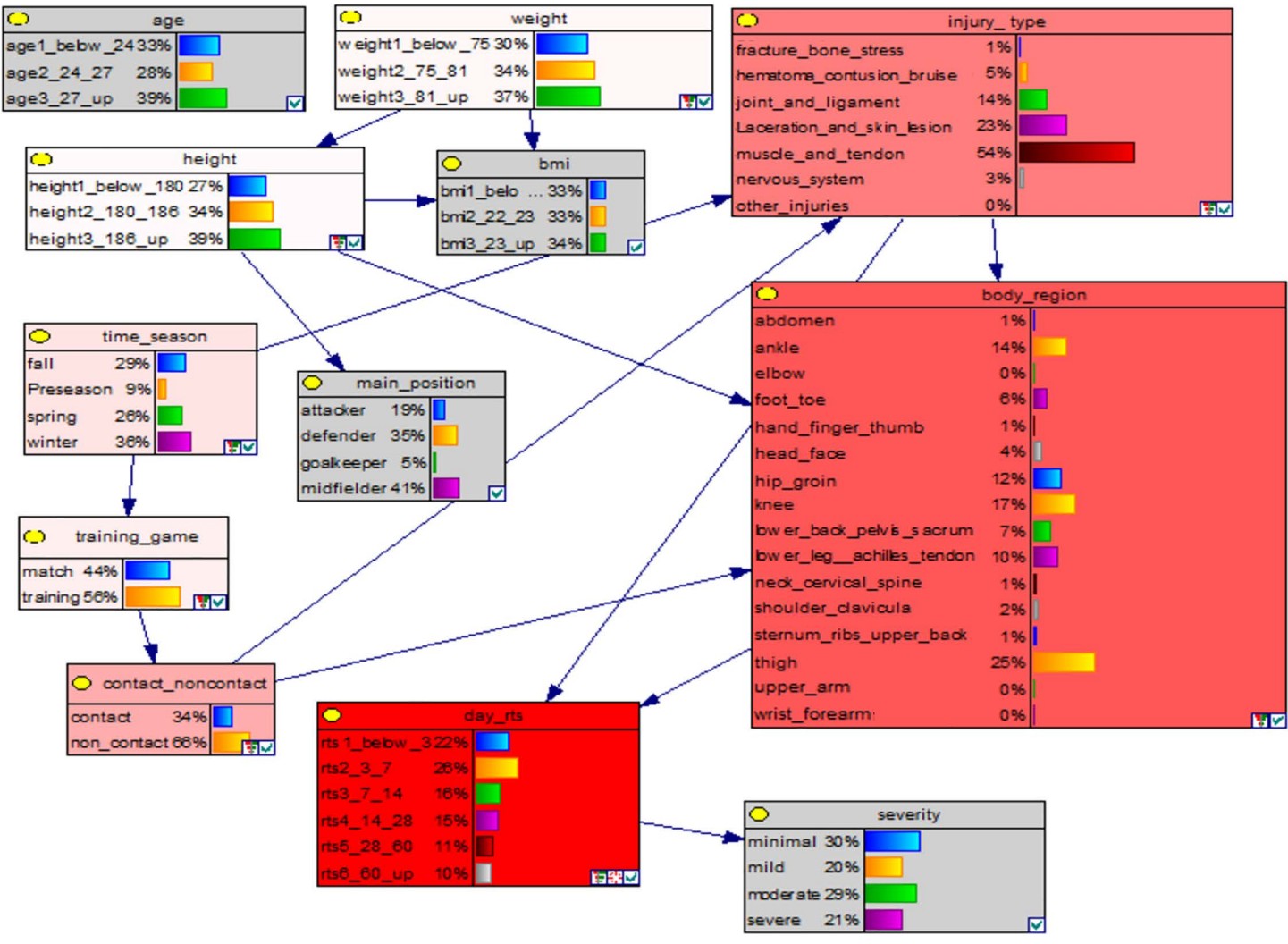

**Fig 3. Feature selection of the Bayesian network.**

The model is more accurate for classifying injuries into 4 categories under *severity* (sensitivity = 0.75 – 1.00), compared to 6 categories *days_rts* (sensitivity = 0.24 – 0.97). This is not surprising because the increase in the number of categories challenged the model to provide a higher accuracy. The model is most accurate in classifying injuries with shorter *days_rts* (days below 3, sensitivity = 0.97) and least accurate in classifying mid-range injuries (days 8–14, sensitivity = 0.24). A possible explanation is that injuries with minimal days of absence have particular injury patterns, for example, they may be upper body injuries and hematoma/contusion/bruise injuries. The model is least accurate for classifying mid-range injuries (8–14 days), possibly because the model lacks information that may differentiate the prognoses. This information may be the extent of tissue damage (e.g., the sub-classification of muscle injury) and the specific location of the injury (e.g., involvement of central tendon injury). When such information becomes available, it is possible to incorporate them into the BN modelling as BNs are quite flexible in accommodating new information. However, the BN will have to be trained again with the new data to see if the model's performance has been improved

## Illustrative application of the BN

Here, we use a hypothetical case study to demonstrate the practicability of using the BN in classifying the days to RTS and the injury severity of a player. We input the player's characteristics and anthropometrics, match information and injury information into the BN constructed earlier. The probability distribution is shown in Fig 4

*A player (age: 25, weight: 77 kg, height: 185 cm, attacker) playing in the German professional football league sprained his ankle ligament in a non-contact injury and pulled out of the preseason training session. The coach would like to know when the player is available for selection.*

The BN indicates the joint multivariate probability distribution: the likelihood of the injury to be minimal is 63%, mild 24%, moderate 12% and severe 1% (Fig 4). The likelihood of RTS below 3 days is 52%, 3–7 days is 33%, 7–14 days is 10%, 14–28 days is 4%, 28–60 days and more

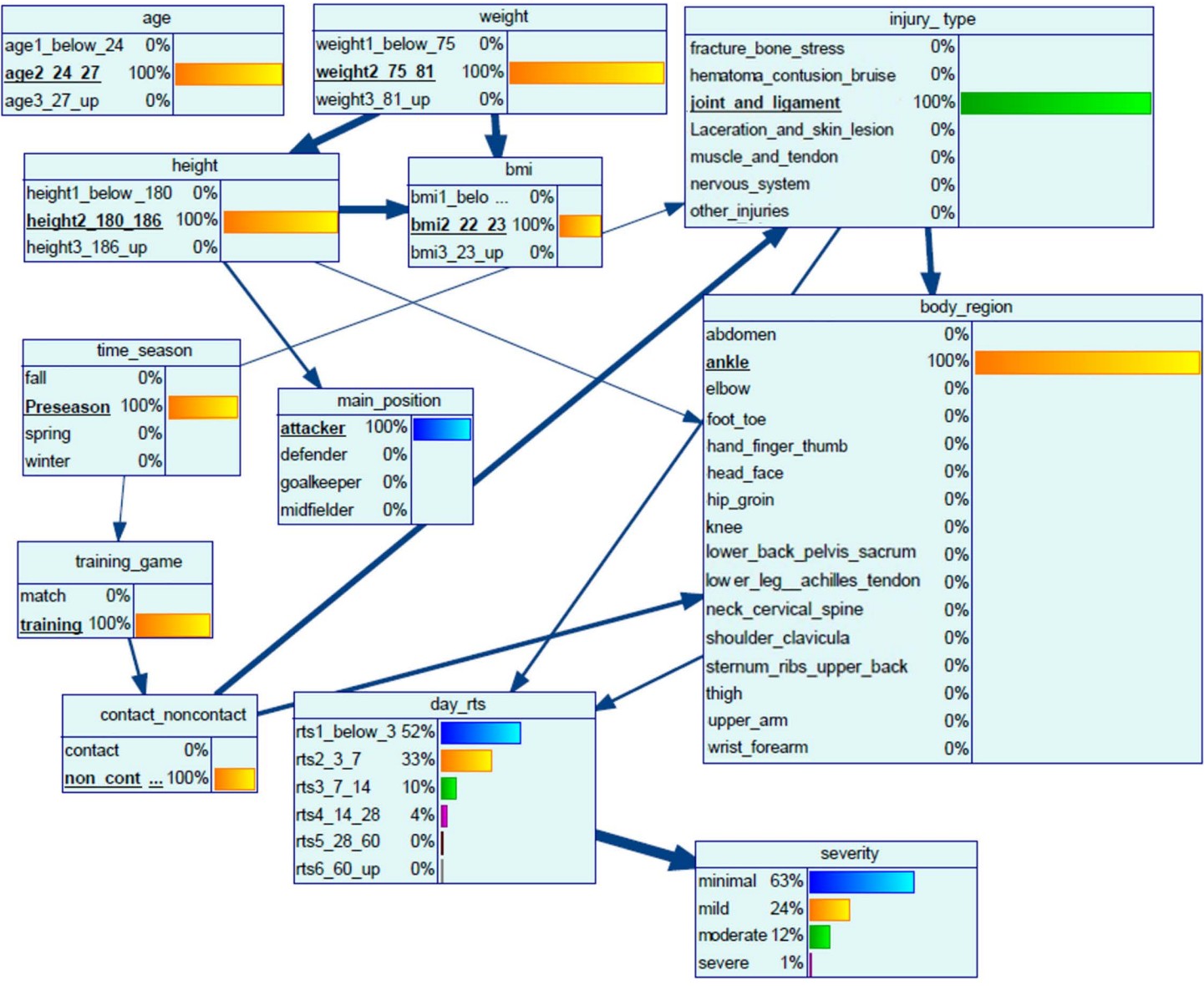

**Fig 4. Bayesian network of the case study.**

than 60 days are less than 1%, respectively. This information, in combination with clinical assessment, may be used to support coaches and players in predicting days to RTS and support team planning by assessing the number of players available for training and competition.

In Fig 4, the thickness of the arcs indicates the strength of influence. Medical staff can use the strength of influence analysis to understand the local relationships among the variables in a network and how they contribute to the posterior marginal probability (i.e., outcome) for each node in the network (Fig 4). The strength of influence calculates the average effect of changing the state of a parent node on the probability distribution of states in the child node[60]. The arcs are normalised, thus the thickest arc indicates the one with the highest influence. From Fig 4, the nature of the injury (contact or non-contact) is related to the injury type and the body region. Injury type and body region are directly associated with the *days_rts*. Medical staff may use *days_rts* in conjunction with the clinical knowledge to project the rehabilitation timeline.

## Practical application and future developments

The software we used for constructing the BN, GeNIe 2.0, could be used as a decision support tool by sports practitioners, including but not limited to coaches, team managers, sports science and medical staff. Once the BN has been constructed, for example, with the use of GeNIe 2.0, the end users do not need any machine-learning knowledge or advanced computer skills to use the BN. Instead, the only input from the sports practitioners is to collect and input the players' characteristics into the BN. The new data from every week or season can be input into the system's database by data scientists (either in-house or third party) to ensure the model is up-to-date. As technology continues to advance, the process of constructing and utilising BN is expected to become easier and more efficient. This increased ease of use may encourage medical staff to invest their efforts in learning and adopting BNs to support their decision-making processes.

When planning for RTS for an athlete, the BN can provide a personalised risk analysis from broad epidemiological data, as exemplified in section "Illustrative application of the BN". While the results of the BN for estimating time to RTS align with existing clinical systems and therefore some practitioners may question the need for a computational decision support tool, it is important to see the potential benefits of integrating BN into sports medical practice in future. Conventional clinical guidelines often apply universally to all individuals regardless of age, sex, sport, or level of play. Consequently, they are not intended to offer an individualised approach to RTS estimation. Given that athletes' diverse needs and circumstances, coupled with the growing emphasis on personalised and precision medicine, a more tailored approach to estimating RTS becomes essential. In this regard, the incorporation of BN can serve as a valuable addition to sports medical practice to improve decision making.

In our example, we explained the use of BN with a RTS scenario and estimated the time to RTS based on a player's basic characteristics. However, it is important to note that this study is a pilot for future studies, and the potential applications of BN extend beyond this specific use case. They can be leveraged for various purposes, such as forecasting expected performance or injury risk in a game by considering factors like personal characteristics, opponent playing style, and recent training and game performance. While in this study, the BN yielded similar results to existing clinical systems, its true benefits lie in its ease of use, efficiency, and the capacity to incorporate a wider range of data into the decision-making process. The BN's ability to identify the strength of influence allows medical staff to address the variables that significantly impact outcomes, a task that cannot be adequately addressed by conventional clinical guidelines alone. By leveraging BN, medical staff may enhance their decision-making processes and provide more tailored and effective care to athletes.

Computational models are more likely to be implemented in applied sports settings if their accuracy, interpretability and functionality fit with the operation framework of a sports organisation[61]. In this case, BN provides an intuitive visualisation of the complex relationship of injuries, which medical staff may understand even with little or no experience in computer analytics. This may increase the model's transparency and may improve the medical staff's trust in the model[62]. In terms of practical applicability, medical staff can use the model to create scenarios that facilitate the evaluation of different management options or the development of rehabilitation protocols for players with various positions or injuries occurring at different times during the season. These scenarios enable the assessment of the combined effects of risk factors[26]. This may enable them to proactively manage injury risk, and in case of injury, make more accurate predictions regarding the days to RTS for individual athletes based on epidemiological evidence. The BN can be modified to suit the specific context as determined by the user and updated with new information when available. In summary, BNs seem promising for modelling the relationships of variables in a complex system and may be further explored to support clinical decision making.

The epidemiological data we used in this study is static; thus, we use a BN to represent the system as a time-aggregated model. However, most systems, including athletes, have been well recognised to change over time[4,63]. To capture the change over time and the feedback loops, a BN can be further extended into a dynamic Bayesian network (DBN)[64]. The DBN is an extension of the BNs and replicates the BN model at discrete points in time (time slices) and captures temporal relationships between the variables. DBNs can represent complex questions, such as how changes in rehabilitation training load affect the time to RTS in athletes with different demographic and anthropometric characteristics. Modifiable variables, such as rehabilitation training exposure, can be collected continuously over time as time-series data[65]. DBNs have been applied in injury prediction[66], medical diagnosis or prognosis [67–69] and are also a promising avenue for further investigation in subsequent injury risk modelling. Specifically, DBNs may provide medical practitioners with a tool to predict the probability of a subsequent injury to occur given time-series data, such as injury history, recovery time and players' physiological state over time. Future work could investigate using Wu et al. (2019)'s approach to analyse scenario-based complex systems[70].

## Limitations

The proposed BN model has limitations that should be considered when interpreting its results. Firstly, it is important to recognise that the model was constructed using data solely from the Bundesliga, and therefore, the generalisability of the results to other leagues or levels of play is limited. Second, the quality of the input variables directly impacts the model's predictive accuracy. In the case of our BN, it was constructed using basic epidemiological data sourced from public databases. While this approach maximises the availability of injury events for the model, it lacks pertinent information related to prognosis, such as diagnosis (e.g., structural damage or functional disorder) and the precise location of the injury (e.g., involvement of central tendon in hamstring injury) [71]. An accurate diagnosis of an injury plays a vital role in determining prognosis[10,72]. For instance, the recovery time to RTS can vary significantly between a minor partial tear in the hamstring muscle (requiring 17 days to RTS) and a moderate partial tear (requiring a longer time of 36 days to RTS). Therefore, the lack of detailed diagnostic information can impact the model's graphical structure and prediction accuracy.

Further studies may consider including more information into the model, such as quality and duration of rehabilitation training, RTS performance, reinjury incidence, the importance of upcoming competition, the remaining contract length, the club's geographical regions and the players' transfer value. Potentially, depending on the purpose of the model, we can

integrate epidemiological data from other larger datasets into the model to enlarge the database to reduce the chance of overfitting and improve result transportability.

Comparing the machine-learned BN with the existing scientific understanding of injury and recovery would also be a valuable future avenue of research. This comparison can provide insights into the performance and predictive capabilities of a complex model that incorporates a wide range of variables in contrast to a simpler model, such as an epidemiological one. By doing so, researchers can evaluate the added value of BN as a decision-support tool.

## Conclusion

This discrete BN provides a decision support tool to help medical staff, coaches and players manage injury. The BN has a high sensitivity, ranging from 0.73 to 1.00 in predicting severity and provides a graphical representation of the investigated interdependencies. Medical staff can use BN to understand the strength of influence of different variables on the outcome and analyse the outcome based on custom RTS scenarios. This information may help medical staff evaluate different injury scenarios and better respond to individual player's rehabilitation and team planning. BNs seem promising for modelling the relationships of variables from multiple sources and can be further explored to support clinical decision making.

## Supporting information

**S1 Fig. Structure of the BN model.**
(PDF)

**S2 Fig. Tornado plots of the BN model.**
(PDF)

**S1 Table. The demographics and the main playing positions of injured players.**
(PDF)

**S2 Table. Detail breakdown of injury.**
(PDF)

**S3 Table. Strength of influence of all arcs in the model.**
(PDF)

## Acknowledgement

Our Bayesian network was built using GeNIe 2.0 Modeler (BayesFusion 2019), available free of charge for academic research and teaching use from https://www.bayesfusion.com/. The authors would also like to thank Tobias Tröß and Abed Hadji for the data collection and injury database management.

## Author contributions

**Conceptualization:** Kate K. Yung, Paul P.Y. Wu, Anne Hecksteden, Tim Meyer.

**Data curation:** Karen aus der Fünten.

**Methodology:** Kate K. Yung, Paul P.Y. Wu, Karen aus der Fünten, Anne Hecksteden, Tim Meyer.

**Supervision:** Paul P.Y. Wu, Karen aus der Fünten, Anne Hecksteden, Tim Meyer.

**Visualization:** Kate K. Yung.

**Writing – original draft:** Kate K. Yung.

**Writing – review & editing:** Kate K. Yung, Paul P.Y. Wu, Karen aus der Fünten, Anne Hecksteden, Tim Meyer.

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
