## [Decision Letter · Decision Letter 0]

10 Dec 2024

PONE-D-24-50228Using a Bayesian network to classify time to return to sport based on football injury epidemiological dataPLOS ONE

Dear Dr. Yung,

Thank you for submitting your manuscript to PLOS ONE. After careful consideration, we feel that it has merit but does not fully meet PLOS ONE’s publication criteria as it currently stands. Therefore, we invite you to submit a revised version of the manuscript that addresses the points raised during the review process.

We look forward to receiving your revised manuscript.

Kind regards,

Benjamin F Mentiplay, PhD

Academic Editor

PLOS ONE

Journal Requirements:

“NO authors have competing interests”

4. In the online submission form, you indicated that [Dataset cannot be shared publicly because it is currently in use for another research project. However, the data would be available on reasonable request.]. 

Reviewers' comments:

Reviewer's Responses to Questions

**Comments to the Author**

1. Is the manuscript technically sound, and do the data support the conclusions?

Reviewer #1: Yes

Reviewer #2: Yes

2. Has the statistical analysis been performed appropriately and rigorously? 

Reviewer #1: Yes

Reviewer #2: Yes

3. Have the authors made all data underlying the findings in their manuscript fully available?

Reviewer #1: Yes

Reviewer #2: Yes

4. Is the manuscript presented in an intelligible fashion and written in standard English?

Reviewer #1: Yes

Reviewer #2: Yes

5. Review Comments to the Author

Reviewer #1: Review of the manuscript titled “Using a Bayesian network to classify time to return to sport based on football injury epidemiological data.”

This manuscript provides an example of the possible utility of using Bayesian Network modelling to determine probabilities for time required to return to play for football athletes. The authors are to be congratulated on providing this example of what might be a particularly useful application of Bayesian statistics in applied sports epidemiology.

I have made the following suggestions for editing.

Line 026: Delete the word “of” after the word “factors” and replace with “related to”.

Line 028: Insert the words “of time to return to play” after the word “estimation”.

Line 028: Delete the word “yet” after the word “and”.

Line 029: Delete the words “often rare in the industry” and replace with “not common in sports medicine”.

Line 034: insert the word “a” after the word “applied”.

Line 035: Insert “. A” after the word “data”

Line 048: It is reasonable to include a description of the method of sensitivity and specificity analysis as similar to the “producers and users “model from cartography. I suggest that for clarity the terms "sensitivity and specificity" are used throughout the manuscript as these terms will be more familiar to sports medicine practitioners?

Line 061: Delete the word “the”.

Line 061: Insert the word “modelling” after the word “network”.

Line 065: Insert the words “defined as occurring” after the word “is”.

Line 065: Insert the words “and play” after the word “training”.

Line 069: Insert the words “support a graduated progression in the athlete's conditioning program” after the word “to”.

Line 069: Delete the word "mitigate” and replace with the words “assist in mitigating”.

Line 070: Insert the words “provision of” after the word “The”.

Line 070: Insert the words “for recovery” after the word timeframe.

Line 077: Insert the word “human physiological” before the word “system”.

Line 092: Delete the words “Due to the complex systems” and start the sentence at the word “Forecasting”.

Line 095: Delete the words “considered” and replace with “included”.

Line 096: Delete the word “these” and replace with “this”.

Line 102: Delete the words “is a lack of tool” and replace with “are no tools”.

Line 103: Delete “on” and replace with “to”.

Line 114: Delete “take charge of” and replace with “undertake the”

Line 129: Insert the word “previously” after the word “been”.

Line 129: Insert the word “a” after the word “such”.

Line 134: Delete the word “In”.

Line 134: There appear to be two Figure 1s? The tornado graph is also labelled as Fig 1 in the supplementary files. I suggest labelling the tornado graph as Fig 4?

Line 135: Delete the word “we”.\

Line 135: Remove the letter “d” from the word “illustrated”.

Line 140: Insert the word “are” after the word “that”.

Line 145: Delete the word “discrete” and replace with the word “descriptive”.

Line 147: Insert the word “the” after the word “of”

Line 147: Delete the word “relevant” and replace with the words “and it’s relevance”.

Line 149: Insert the word “also” after the word “it’s”.

Line 160: Delete the word “brought” and replace with “made”.

Line 170: Add an “s” to the word “provide”.

Line 179: Delete “combination of both of them” and replace with “combinations of these”.

Line 189: Delete “use” and replace with “utility”.

Line 193: Delete the word “a”.

Line 253: Insert the words “competition or” after the word “in”.

Line 259: Insert the word “Author KY established” and delete “helped establish”.

Line 260: Delete “our” and replace with “the”.

Line 304: As suggested previously, I suggest consider explaining the similarity between these and sensitivity and specificity and using these terms instead as these will be more easily understood by health readers?

Line 391: Delete “’ eyes”.

Line 482: Delete “yield” and replace with “yielded”.

Line 484: Delete the word “Moreover” and replace with the word “The”.

Line 507: Insert the word “The” before the second use of the word “DBN”.

Line 508: Insert the word “the” after the word “of”.

Line 528: It may be worthwhile extending the discussion to include the possibility of extending the DBN model to examine the possibility of modelling subsequent injury risk?

In the “Supplementary Material Table 3” I suggest that a short explanation of Ave, Max, +Weighted is added to the legend.

Reviewer #2: General Comments

This paper demonstrated how a Bayesian network model can be used to estimate time to RTS in professional male German footballers. Bayesian networks are very useful in this context, and I commend the authors for using this model, which I think underused in sport, outside of the anti-doping context. Except for the abstract and background, which can be written more clearly, the methods and results were easy to understand. The discussion is relevant, provides a concrete example and does not attempt to overplay the usefulness of the model, which is to augment/guide clinical decision making.

Background: The background can be written more clearly and concisely. Several phrases can be removed without losing meaning. Some paragraphs require some extra context to make them easier to interpret. I’ve provided some examples how various statements can be written more concisely to improve readability.

Specific Comments

ABSTRACT

The abstract can be written more concisely in several areas without losing meaning. I have provided examples in each section.

Objective:

Original: “This study aims to demonstrate the functions of a Bayesian network by the use of a set of basic epidemiological data.”

Revised: “This study uses epidemiological data to demonstrate a Bayesian Network.”

Methods:

Original: “To exemplify the use of Bayesian network in sports medicine, such as providing an individualised estimation time to RTS for individual players, we applied Bayesian network to a set of basic epidemiological data. Bayesian network was used as a decision support tool to model the epidemiological data and to integrate clinical data, non-clinical factors and expert knowledge. Specifically, we used the Bayesian network to capture the interaction between variables in order to, 1) classify days to RTS and 2) injury severity (minimal, mild, moderate and severe).”

Revised: “We applied a Bayesian network that integrated clinical, non-clinical factors, and expert knowledge to classify time day to RTS and injury severity (minimal, mild, moderate and severe) for individual players”

Original: “Retrospective injury data of 3374 player seasons and 6143 time-loss injuries from seven seasons of the professional German football league (Bundesliga, 2014/2015 through 2020/2021) were collected from public databases and media resources. A total of twelve variables from three main categories (player’s characteristics and anthropometrics, match information and injury information) were included. The key response variables are 1) days to RTS (1-3, 4-7, 8-14, 15-28, 29-60, >60 , and 2) severity (minimal, mild, moderate and severe). As there are more than two categories, producer’s and user’s accuracy was used to reflect the sensitivity and specificity of the model. The producer’s accuracy of the model for days to RTS ranges from 0.24 to 0.97, while for severity categories range from 0.73 to 1.00. The user’s accuracy of the model for days to RTS ranges from 0.52 to 0.83, while for severity categories, it ranges from 0.67 to 1.00.”

Revised: “Retrospective injury data of 3374 player seasons and 6143 time-loss injuries from seven seasons of the professional German football league (Bundesliga, 2014/2015 through 2020/2021) were collected from public databases and media resources. A total of twelve variables from three categories (player’s characteristics and anthropometrics, match information and injury information) were included. The response variables were 1) days to RTS (1-3, 4-7, 8-14, 15-28, 29-60, >60, and 2) severity (minimal, mild, moderate, and severe). As there were more than two categories, the producer’s and user’s accuracy were used to reflect the sensitivity and specificity of the model. The producer’s accuracy of the model for days to RTS was 0.24-0.97, while for severity categories was 0.73-1.00. The user’s accuracy of the model for days to RTS was 0.52-0.83, while for severity categories, it was 0.67-1.00.”

Conclusion

Original: “The Bayesian network can help to capture different types of data to model the probability of an outcome, such as days to return to sports. In our study, the result from the BN may support coaches and players in predicting days to RTS given an injury, 2) support team planning via assessment of scenarios based on player’s characteristics and injury risk and 3) provide evidence-based support of understanding relationships between factors and RTS. This study shows the key functions and applications of the Bayesian network in RTS, and we suggest further experimenting and developing the Bayesian network into a decision-supporting aid.”

Comment: You have abbreviated Bayesian network as BN for the first time in the conclusion. If you are going to do this, then abbreviate at first use BN from then on.

Revised: “The Bayesian network can help to integrate different data types to model the probability of an outcome, such as days to return to sport. In our study, the Bayesian network may support coaches and players in predicting days to RTS given an injury, 2) support team planning via assessment of scenarios based on players’ characteristics and injury risk, and 3) support an evidence-informed understanding of the relationships between injury risk factors and RTS. This study demonstrates the how a Bayesian network may aid clinical decision making for RTS.”

BACKGROUND

Ln 72-75 – Original: The test below can be written more concisely, as demonstrated:

“There is a growing acknowledgement that athletes operate within a complex system and are subject to the influences of a multitude of variables(4, 5). These factors include previous injury history, current injury, body mass index, playing positions, sociological factors, psychological status, and the nature of the sports event(4).”

Ln 72-75 – Revised: “Athletes operate within a complex system and are influenced by factors such injury history, current injury, body mass index, playing positions, sociological factors, psychological status, and the nature of the sports event(4).”

Ln 77-81 – Original: “The human body's complexity and the system's highly intricate nature further complicate the task. Understanding these interactions and their impact on injuries becomes crucial for providing accurate prognoses and projecting a range of the likely RTS time (e.g., there is a 75% chance for the player to return between 4-5 weeks). This could help medical staff and athletes to prepare for both ends of the scenario.”

I think the above an be written more clearly. Suggestion below:

Ln 77-81 – Revised: “Understanding the interactions between body systems and injury risk is crucial to provide accurate injury prognoses and project an athlete’s likely RTS time range (e.g., there is a 75% chance for the player to return between 4-5 weeks). Such RTS projections could help medical staff and athletes to prepare RTS programmes based on both the lower and upper projection limits.”

Ln 82-91 – Original: “Large-scale epidemiological studies have made promising strides in providing valuable insights into the expected time to RTS for major injuries in football at the population level(6-9). These studies describe the risk factors for a disease or injury and the extent of the problem(10). Medical staff can use them as a starting point for estimating the time to RTS(11). This approach aligns with the “anchor and adjust” strategy(12), which can be used to optimise predictive accuracy, especially when making decisions in uncertain scenarios (e.g., injury). However, typical epidemiological studies are intended to offer correlations and data based on groups to reflect population-level insights and trends. While this data is valuable for highlighting general patterns and insights, a tool to translate population-level data into individual-level assessments and allow for personalised estimations would be helpful.”

Ln 82-91 – Revised: “Large-scale epidemiological studies have estimated the expected RTS time for major injuries in football at the population level accounting for associated injury risk factors(6-9). Medical staff can then use an “anchor and adjust strategy” to optimise improve predictive accuracy, thus better estimating RTS times. While typical epidemiological studies aim to estimate population-level effect and trends, tools to translate population-level data into individual-level estimations would be helpful.”

Ln 92-96 – Original: “Due to the complex systems, forecasting the time to RTS based on individual characteristics can be challenging. There are several reasons why this process is not straightforward. First, personal characteristics and other factors, such as age and playing position, are recommended to be considered in the Strategic Assessment of Risk and Risk Tolerance framework(13).”

Comment: The phrase “complex systems” is vague. What complex systems are you referring to here. I think this statement can be removed without losing meaning, as in the example below:

Ln 92-96 – Revised: “Forecasting the time to RTS based on individual characteristics can be challenging for several reasons. First, the Strategic Assessment of Risk and Risk Tolerance framework recommends considering personal characteristics and factors, such as age and playing position (13).”

Ln 102: Change “There is a lack of tool” to “There is a lack of TOOLS”

Ln 107: Add making: “Decision MAKING tool”

Ln 114: Add the “s” to “setting”. “In practical SETTINGS”

Ln 123: “This includes but is not limited to understanding” should be, “this includes, but is not limited to”. Please add the missing commas.

METHODS

Ln 140: Add the missing “are” to complete the sentence: “there are merits of using the discrete Bayesian network that ARE worth discussing.”

Ln-193-196: This section is not needed. Your section subheadings provide this information.

2.2 – Participants: Was there any consideration for time played? For example, if a player came on for 5 minutes at the end of a game vs. played a full 90 minutes. To some extent, their exposure (time) to injury risk factors during match play is greater and probably needs to be considered. It is also unclear to me from the participants section if training data was collected or not. I presume not if the BN was built from open-source data. Can you state this more clearly?

DISCUSSION

Ln 413-424: It could be worth discussing how the model can be improved in your final sentence. You have suggested that the contextual information about the extent of the tissue damage and injury location is missing and that is a limitation of the model or reason for reduced accuracy. However, could this information be included in an updated version of the model? If so, I suggest make this clear for the reader as Bayesian network models are quite flexible and can accommodate new information.

4.2 – Illustrative Example

This section is nice as it provides a concrete example of how the BN can be used.

I appreciate this is a static example based built on historic data, but the BN is deigned to be used prospectively. Can you model accommodate validation information/model updating? For example, let’s pretend the medical team use the BN to predict the severity and days RTS from a non-contact injury (before assessment by the medical team). The model might come up with a mild injury, 3-7 days until RTS. However, when the medical staff assess the player, they think the injury is severe and the player’s recovery time is much longer, say 28-60 days. Clearly the model is wrong.

In the context of a dynamic Bayesian network that can be updated with new data to improve model accuracy, is there any process for the user to refine the model once the clinical outcome is known? Obviously, more data helps, but could some form of user input or reinforcement could be useful to refine the model? For example, if the user can tell the model that it has misclassified the player, then that could be useful to refine future predictions. This is also a consideration in section 4.3.

6. PLOS authors have the option to publish the peer review history of their article (what does this mean? ). If published, this will include your full peer review and any attached files.

**Do you want your identity to be public for this peer review?** For information about this choice, including consent withdrawal, please see our Privacy Policy .

Reviewer #1: No

Reviewer #2: No

---

## [Author Response · Author response to Decision Letter 0]

22 Jan 2025

Please see attachment for the respond to reviewers.

---

## [Decision Letter · Decision Letter 1]

13 Feb 2025

Using a Bayesian network to classify time to return to sport based on football injury epidemiological data

PONE-D-24-50228R1

Dear Dr. Yung,

We’re pleased to inform you that your manuscript has been judged scientifically suitable for publication and will be formally accepted for publication once it meets all outstanding technical requirements.

Kind regards,

Benjamin F Mentiplay, PhD

Academic Editor

PLOS ONE

Additional Editor Comments (optional):

Reviewers' comments:

Reviewer's Responses to Questions

**Comments to the Author**

1. If the authors have adequately addressed your comments raised in a previous round of review and you feel that this manuscript is now acceptable for publication, you may indicate that here to bypass the “Comments to the Author” section, enter your conflict of interest statement in the “Confidential to Editor” section, and submit your "Accept" recommendation.

Reviewer #1: All comments have been addressed

Reviewer #2: All comments have been addressed

2. Is the manuscript technically sound, and do the data support the conclusions?

Reviewer #1: Yes

Reviewer #2: Yes

3. Has the statistical analysis been performed appropriately and rigorously? 

Reviewer #1: Yes

Reviewer #2: Yes

4. Have the authors made all data underlying the findings in their manuscript fully available?

Reviewer #1: Yes

Reviewer #2: Yes

5. Is the manuscript presented in an intelligible fashion and written in standard English?

Reviewer #1: Yes

Reviewer #2: Yes

6. Review Comments to the Author

Reviewer #1: Line 451 custom RTS scenarios (section ection 4.Illustrative application of the BN2), understanding the

Suggest replace the word “section with the words “in the section titled”

Line 452 most influential factors (section Sensitivity analysis, feature selection and strength of

Suggest replace the word “section with the words “in the section titled”

Line 476 injury). When such information become available, it is possible to incorporate them into the

Suggest changing the word “become” to the word “becomes”.

Reviewer #2: (No Response)

7. PLOS authors have the option to publish the peer review history of their article (what does this mean? ). If published, this will include your full peer review and any attached files.

**Do you want your identity to be public for this peer review?** For information about this choice, including consent withdrawal, please see our Privacy Policy .

Reviewer #1: **Yes: ** Gordon Waddington

Reviewer #2: **Yes: ** Andrew Govus

---

## [Editor Report · Acceptance letter]

PONE-D-24-50228R1

PLOS ONE

Dear Dr. Yung,

I'm pleased to inform you that your manuscript has been deemed suitable for publication in PLOS ONE. Congratulations! Your manuscript is now being handed over to our production team.

Kind regards,

on behalf of

Dr. Benjamin F Mentiplay

Academic Editor

PLOS ONE